Multi-decade biomass dynamics in an old-growth hemlock-northern hardwood forest, Michigan, USA

Woods Kerry D. kwoods@bennington.edu
Natural Sciences, Bennington College , Bennington, VT , USA
Shugart Herman
Electronic publication date: 2014 Sep 30
Publication date: 2014
Volume: 2
Electronic Location ID: e598
Received 2014 Jul 16; Accepted 2014 Sep 6
Copyright: © 2014 Woods
Copyright year: 2014
Copyright holder: Woods
License: This is an open access article distributed under the terms of the Creative Commons Attribution License, which permits unrestricted use, distribution, reproduction and adaptation in any medium and for any purpose provided that it is properly attributed. For attribution, the original author(s), title, publication source (PeerJ) and either DOI or URL of the article must be cited.
License URL: https://creativecommons.org/licenses/by/4.0/

Keywords: Long-term studies, Old-growth forest, Forest carbon pools, Northern hardwood forest, Temperate forest

Funding: National Science Foundation DEB-9221157 Andrew W. Mellon Foundation Huron Mountain Wildlife Foundation Bennington College This work has been supported by funding from: The National Science Foundation (DEB-9221157), The Andrew W. Mellon Foundation, The Huron Mountain Wildlife Foundation, and Bennington College. The funders had no role in study design, data collection and analysis, decision to publish, or preparation of the manuscript.

==============================
Trends in living aboveground biomass and inputs to the pool of coarse woody debris (CWD) in an undisturbed, old-growth hemlock-northern hardwood forest in northern MI were estimated from multi-decade observations of permanent plots. Growth and demographic data from seven plot censuses over 47 years (1962–2009), combined with one-time measurement of CWD pools, help assess biomass/carbon status of this landscape. Are trends consistent with traditional notions of late-successional forests as equilibrial ecosystems? Specifically, do biomass pools and CWD inputs show consistent long-term trends and relationships, and can living and dead biomass pools and trends be related to forest composition and history? Aboveground living biomass densities, estimated using standard allometric relationships, range from 360–450 Mg/ha among sampled stands and types; these values are among the highest recorded for northeastern North American forests. Biomass densities showed significant decade-scale variation, but no consistent trends over the full study period (one stand, originating following an 1830 fire, showed an aggrading trend during the first 25 years of the study). Even though total above-ground biomass pools are neither increasing nor decreasing, they have been increasingly dominated, over the full study period, by very large (>70 cm dbh) stems and by the most shade-tolerant species (Acer saccharum and Tsuga canadensis).

CWD pools measured in 2007 averaged 151 m3/ha, with highest values in Acer-dominated stands. Snag densities averaged 27/ha, but varied nearly ten-fold with canopy composition (highest in Tsuga-dominated stands, lowest in Acer-dominated); snags constituted 10–50% of CWD biomass. Annualized CWD inputs from tree mortality over the full study period averaged 1.9–3.2 Mg/ha/yr, depending on stand and species composition. CWD input rates tended to increase over the course of the study. Input rates may be expected to increase over longer-term observations because, (a) living biomass is increasingly dominated by very large trees whose dead trunks have longer residence time in the CWD pool, and (b) infrequent major disturbances, thought to be important in the dynamics of these forests, have not occurred during the study period but would be expected to produce major, episodic pulses in CWD input.

Few fragments of old-growth cool-temperate forests remain, but such forests can constitute a very large carbon pool on a per-area basis. The carbon sink/source status of these forests remains unclear. While aboveground living biomass at this study site shows no strong aggrading or declining trend over the last half-century, this remains a modest span in the innate time-scale of late-successional forest. The effects of rare disturbances, long-term shifts in composition and size structure, and changes in soil carbon and CWD pools may all influence long-term carbon status.

Introduction

Old-growth temperate forests include stands with the highest known densities of living biomass (Keith, Mackey & Lindenmayer, 2009) and large pools of persistent dead biomass add substantially to total carbon reservoirs. In typical old-growth forests, large individual trees contribute disproportionately to total biomass in the highest-biomass stands (Brown, Schroeder & Birdsey, 1997; Keith, Mackey & Lindenmayer, 2009; Keeton et al., 2011). However, these generalizations are based on a relatively few studies with biomass estimates from direct measurement of trees—fewer than a dozen from northeastern North America (see below)—and nearly all of these are based on one-time measurements. Consequently, although several recent authors have suggested that old-growth forests can act as carbon sinks as well as important carbon reservoirs (Keeton et al., 2010; Keeton et al., 2011), the temporal dynamics of living or dead biomass components have been inferred from modeling of forest growth or from space-for-time substitution. Models of old-growth systems often assume, explicitly or implicitly, that community and ecosystem processes are in steady state, but the long-term data-sets necessary to assess such assumptions are rare. A few long-term studies and some simulation models suggest that, in fact, composition may not be steady-state even in old-growth stands (Pacala, Canham & Ribbens, 1996; Woods, 2004; Woods, 2007). Consequently, it is important to assess whether ecosystem properties like biomass density also show significant general or predictable trends in old-growth forests.

Using data from permanent plots with multiple remeasurements over 47 years, I calculate aboveground biomass dynamics in old-growth hemlock-northern hardwood forests in northern Michigan, US. Combined with a one-time measurement of the standing CWD pool, these data allow assessment of behavior of both living and dead biomass and carbon pools in a cool-temperate old-growth forest.

Coarse woody debris in (CWD) in hemlock-northern hardwood forests plays an important role in mediating nutrient and carbon cycling (Liu et al., 2006; Gough et al., 2007), in providing substrate for plant germination and seedling growth (Marx & Walters, 2008; Bolton & D’Amato, 2011), and in structuring habitat for many other species (Harmon et al., 1986; D’Amato, Orwig & Foster, 2009). Past estimates of CWD density for old-growth hemlock-northern hardwood forests are generally based on one-time inventories, but variation in rates of CWD input could have far-reaching consequences beyond effects on carbon sequestration. In old-growth forests, where large logs can persist on the forest floor for decades, historical variation in biomass and CWD dynamics could leave long-persistent signatures. One-time measurements may, then, be influenced by historical legacies of rare disturbances and not reflect stable conditions. Mesic forests of the upper Great Lakes region appear to be structured predominantly by wind disturbances of varying intensity and frequency (Brewer & Merritt, 1978; Canham & Loucks, 1984; Woods, 2004). If biomass and CWD inputs are strongly influenced by rare, severe wind events, they may be significantly altered by changing frequency and severity of such events due to predicted climate changes.

Specifically, I ask:

- Do living biomass pools show consistent trends at decadal time scales?

- How do inputs of CWD through tree mortality vary over time at decadal scales?

- Are historical CWD input rates, consistent with current CWD constituting a steady-state pool?

- Do living and dead biomass pools and trends vary with forest composition and history in consistent ways?

Methods

Study-sites and field sampling

Analyses are based on repeated measurements of nineteen 0.2 acre (0.081 ha) circular permanent plots established in 1962 by Eric Bourdo (Ford Forestry Center, Michigan Technological University) within an approximately 5000 ha tract of old-growth forest near Big Bay, MI and within 3 km of the Lake Superior shoreline (approximately 46°52′N 87°54′W). This tract has been owned by the Huron Mt. Club for over 120 years—since before any logging activity in the immediate vicinity—and has been free from any active management since acquisition.

The Bourdo plots are distributed over a range of habitats and community compositions (Fig. 1). Substrate ranges from acidic, rocky soils developed on thin glacial drift over pre-Cambrian granites to coarse sandy soils, to soils on fine-textured alluvial deposits with relatively high cation availability. Canopy composition ranges from strong hardwood (Acer saccharum Marshall, Tilia americana L.) dominance on high-cation soils to varying mixes of A. saccharum and Tsuga canadensis (L.) Carrière on other soils (Woods, 2000). Three plots are in a Tsuga-dominated stand originating following a fire about 1830 (prior to significant European-American presence in the area); all other sampled stands are multi-aged with individual stems exceeding 300 yr old and little evidence of strong cohort structuring (K Woods, 1994, unpublished increment core data). All live trees greater than 5 inches in diameter at breast height were measured and marked in 1962 and Bourdo and colleagues remeasured live trees in 1967.

Figure 1 Study area and examples of forest types.

(A) The Huron Mts. study area with local topography (Lake Superior surface elevation is 180 m asl and the highest areas in the inset are about 550 m asl), and examples of plot groupings used in analyses; (B) mixed conifer-hardwood stands; (C) plots burned in 1830 with Tsuga dominance; (D) Tsuga-dominated; and (E) Acer-dominated hardwoods (all photos by Kerry Woods).

In 1989, I remeasured the Bourdo plots, mapping all stems >5 cm dbh for the entire plot and >2 cm for a central sub-plot of 8 m radius. Painted stem-numbers were still visible, allowing tracking of individual trees from earlier measurements. Subsequently, plots have been remeasured every five years through 2009, documenting growth, mortality, and recruitment. In all measurements, dead trees were recorded as dead standing, tipped up, or broken (with height of break).

In 2007 I censused CWD on the Bourdo plots. For any piece of down wood partially within the plot and >10 cm diameter at any point, we measured end diameters and mapped both ends of the piece, recording polar coordinates with a LaserCraft Contour XLRic. Bent or branched CWD segments were subdivided into relatively straight, unbranched segments for mapping. For CWD segments resting on slopes or not resting flat on the ground, we recorded a height (z-coordinate) as well. All CWD segments were, where possible, identified to species and assigned to an identified stem. We measured heights of stumps and diameter at top of stump (approximating top diameter where it was too high to measure directly) and at ground surface (approximating an average diameter where ground surface was irregular). If there was significant basal flare, the stump was divided into segments to reduce non-linearity of taper. For snags (standing dead trees with bole intact to diameter >10 cm), we recorded height and diameter at breast height as well as basal diameter.

All CWD segments were scored according to a five-point decay classification similar to that used by other researchers (Angers et al., 2005): Class 1, bark intact and tight; Class 2, significant loss of bark, but wood intact or only slightly decayed; Class 3, outer layers of wood clearly soft and decaying, but log structurally intact, rigid; Class 4, decayed and soft throughout, losing structural integrity (‘slumping’) but still mostly round in cross-section; Class 5, very soft, disintegrating, often moss-covered, no longer round in cross-section (Fig. 2). For the last category, diameters were estimated by averaging approximate vertical and horizontal diameters.

Figure 2 Examples of decay classes.

Examples of decay classes used in classifying CWD: (A) recently fallen Acer saccharum, class 1; (B) Acer saccharum log in class 2; (C) Acer saccharum log in class 3 or 4; (D) Tilia americana log in class 5 (all photos by Kerry Woods).

Analyses

Estimating living biomass and CWD inputs

Above-ground living biomass was estimated for all trees >12.5 cm dbh (to conform with the 5 inch minimum dbh recorded in 1962 and 1967), using published dbh-based allometric equations. Equations were selected from Jenkins et al. (2004) on criteria including: location of development, favoring equations developed for the upper Great Lakes region; size range, favoring relationships based on samples including large trees; and size of sample used to develop relationships (see Table S1 for equations used and original sources).

It is not possible to assess either precision or accuracy of resulting estimates, but significant error and bias are probably inescapable. For some species, equations were not available for the upper Great Lakes region. No available allometric equations were fitted using trees larger than about 70 cm dbh. For all major canopy species in the current data-set, individuals of >70 cm dbh are relatively common, with maximum sizes around 100 cm dbh, and large trees account for substantial fractions of total biomass. Where multiple appropriate equations were available, I compared resulting estimates; estimates for trees up to 50–60 cm dbh were generally similar among equations but, for larger stems, they diverged substantially—by as much as 30% for a 90 cm stem.

Inputs to the CWD pool were estimated using equations for bole biomass only, applied to trees that died during the interval between two measurements. In most cases, CWD input estimates were based on dbh from the last measurement before the tree’s death, so any growth after that measurement and before death would be small (annual diameter increments for canopy trees in this study average around 1.0–1.5 mm/yr depending on species and interval, and growth rates are typically reduced in the years immediately preceding death (Woods, 2000)). However, for the 22-year interval between 1967 and 1989, I used diameter measurements of standing dead trees taken in 1989 where loss of bark and decay were not significant. As much as 30% of above-ground living biomass, in large trees, can be in branches, twigs, and foliage. However, only the very largest branches would contribute significantly to CWD as measured here, and allometric equations typically do not allow differentiation of branches by size. The effects of using dbh for up to several years prior to death and of using bole biomass only will result in some systematic underestimation of CWD input rates. CWD inputs are presented as annualized rates, evenly distributed within each interval between measurements.

Estimating CWD volume and mass

Volume of individual CWD segments was estimated by treating them as regular conic frusta; the height (length) of the frustum was derived from the mapped polar coordinates of the end-points (using three-dimensional coordinates where appropriate). For CWD segments lying only partially within the plot, I calculated the length within the plot perimeter and estimated diameter at that point assuming linear taper between measured ends; volume was calculated for the portion within the plot only. Stumps were also treated as frusta (or multiple frusta where flare led to sectioned measurements). Snag volume was estimated using similar approaches unless appropriate dbh-based allometric equations for bole volume were available (Table S1). CWD volume was converted to dry biomass using taxon-specific relationships between density and decay class from Liu et al. (2006), which uses decay classes nearly identical to those used here (for unidentifiable fragments in decay classes 3 to 5, values for ‘all species’ were used).

Plot groupings

For comparisons across stands of different history and composition, I used plot groupings based on location, habitat, and canopy composition dominance (see Woods, 2000). The Acer group includes plots in two localities with fine-textured, high-cation soils and near-complete hardwood dominance and with maximum ages >300 years. The Tsuga group includes two plots in a single stand with high hemlock dominance and apparent maximum age of about 300 years (K Woods, 1993, unpublished increment core data). Six plots on level, coarse, deep sandy soils, with upper canopy dominated by hardwoods but significant subcanopy and codominant Tsuga component were initially assigned to a ‘mixed flat’ group and three plots in areas of shallower soils on moderate slopes and with mixed hardwood-Tsuga canopy were treated as a ‘mixed slope’. However, all results and trends were very similar for these two groups, and they are pooled here as a single ‘mixed’ plot group. Finally, three plots in the 1830 burn area (slight to moderate slopes, and strong Tsuga dominance) were assigned to the ‘burn’ group.

Statistical approaches

Changes in plot-level characteristics over the full study period were assessed using paired t-tests comparing starting and ending values with appropriate transformations. Differences among compositional plot groupings were assessed using non-parametric Kruskal–Wallis tests with post hoc Mann–Whitney pair-wise tests with Bonferroni corrections.

Results

Aboveground living biomass

Aboveground living biomass densities at the most recent measurement date ranged from about 360 to 450 Mg/ha within plot groups (Table S2 for values by species and plot). Trends over 47 years were not consistent over time or among groups (Table 1 and Fig. 3). In the Acer and mixed groups total change was less than 10% of initial values, although the mixed group reached a peak biomass density in 1994 nearly 20% greater than the initial value. Both Tsuga-dominated groups (‘Tsuga’ and ‘burn’), however, showed more consistent and substantial overall increases in biomass, although biomass density in the ‘burn’ group decreased by about 10% (after a ca. 25% increase) in the last decade of the study. Averaged over all plots, biomass density increased from 358 Mg/ha in 1962 to 378 Mg/ha in 2009, but this change was not statistsically significant (paired t-test, p > 0.05).

Figure 3 Trends in above-ground living biomass.

Trends in above-ground living biomass over time and among composition-based plot groups. Overall average biomass density increased slightly, from 358 Mg/ha to 378 Mg/ha from 1962 to 2009. All plot groups show changes in direction of change over time. Highest values, for strongly Acer and Tsuga- dominated stands, are among the highest reported for eastern North America.

Table 1 Total aboveground biomass, stems >12.5 cm dbh (Mg/ha).

	Year	
Plot group	1962	1967	1989	1994	1999	2004	2009	
Acer (n = 5)	434.2	404.7	438.8	436.7	433.3	434.0	442.4	
Tsuga (n = 2)	377.3	363.0	425.0	435.2	448.2	444.8	452.9	
Mixed (n = 9)	331.2	341.8	369.5	377.9	368.9	348.9	339.4	
Burn (n = 3)	296.5	305.5	364.0	363.9	368.3	350.3	337.0	

The proportion of total biomass in stems >70 cm dbh, over all plots, increased from 6% in 1962 to 14% in 2009 (paired t-test, arcsin conversion, p < 0.01) (Fig. 4A). Acer plots were most strongly dominated by very large trees over the entire study period (Fig. 5), but proportional representation peaked in 1989. Tsuga and mixed plots saw more continuous increases in big-tree fractions, while large stems never exceeded 5% of biomass in the ‘burn’ plots (all individuals >70 cm dbh were Populus grandidentata Michx., and all died by 2004).

Figure 4 Cumulative distribution of biomass and CWD with respect to tree size.

(A) Cumulative curves (from largest to smallest stems) for total biomass and stem density are higher for 2009 over nearly the full span of stem diameters. Large trees increasingly dominate both living biomass and stem density over the study period, but (B) CWD inputs through mortality do not show similar trends. CWD input for the first 27 years of the study does not differ from curve for the last twenty years; numerically, mid-size canopy trees dominate inputs in the earlier period, but smaller stems dominate in the later period.

Figure 5 Proportion of living biomass in large trees.

Proportion of living biomass in large trees (>70 cm dbh) increases over time for all plot groups except the ‘burn’ group. Mortality of early-successional species accounts for the loss of large trees in the burn group after 1999.

In all plot groups where Tsuga had significant presence, it increased as a proportion of total biomass (Fig. 6 and Table S2). Acer biomass changes varied in absolute values, but its proportional importance decreased except in the ‘Acer’ plot group. Betula alleghaniensis Britton and Tilia americana biomass showed varying trends, but increases were generally modest while declines were substantial in some plot groups. Eleven other species (Woods, 2000, Table S2) reached diameters of 12.5 cm, but, in total, represented no more than 8% of total biomass except in the ‘burn’ group, where they were 17% of total biomass in 1962 (Populus grandidentata constituted more than half of this portion). In all plot groups, representation of minor species decreased consistently and substantially over the course of the study as large stems died without replacement.

Figure 6 Living biomass by species over time.

In all plot groups, the most shade-tolerant canopy species (Tsuga canadensis and Acer saccharum) increasingly dominate biomass pools over the course of the study. Tsuga, where present, shows the largest proportional increase, while other species show generally decreasing biomass contributions.

Coarse woody debris

Coarse woody debris pools in 2007, including snags, averaged 151 m3/ha (range among plot groups, 119–178 m3/ha) and 46.0 Mg/ha (38–52 Mg/ha) (Table 2, Fig. 7 and Table S3). CWD densities were significantly higher in Acer-dominated stands, but were not otherwise different among plot groups (p > 0.05, Kruskal–Wallis test, post hoc Mann–Whitney pair-wise tests with Bonferroni correction). The proportion of CWD in the first two decay classes was substantially lower in the Acer group (31% by mass) compared to other groups (47–61%), and highest for the ‘burn’ group. Snag densities ranged from 9.9/ha (‘Acer’ group) to 98.8/ha (‘burn’) with an overall average of 27.3/ha. Snags constituted 10–20% of total CWD biomass for mixed and hardwood dominated plot groups, but 49% for the ‘Tsuga’ group and 99% for the burn group (Table S3).

Figure 7 CWD distribution by decay class and plot group.

Coarse woody debris (CWD) pools in 2007 were highest in Acer-dominated plots in both total mass and voume. Acer plot group CWD pools—composed entirely of hardwood species—are proportionally more dominated by more fully decayed material.

Table 2 Coarse woody debris, average values by plot group, 2007.

	Decay class	
Plot group	I	II	III	IV	V	Total	
I. CWD volume (m3/ha)							
Acer (n = 5)	10.0	29.8	52.2	50.7	35.6	178.3	
Tsuga (n = 2)	18.5	34.0	62.3	19.8	8.6	143.2	
Mixed (n = 9)	6.4	53.0	29.6	35.5	14.1	138.6	
Burn (n = 3)	26.5	45.9	22.8	17.1	37.8	150.1	
II. CWD dry biomass (Mg/ha)							
Acer (n = 5)	4.7	11.4	15.5	11.9	8.5	52.1	
Tsuga (n = 2)	8.0	11.7	16.8	4.0	1.9	42.4	
Mixed (n = 9)	2.8	20.5	8.7	7.8	3.3	43.0	
Burn (n = 3)	11.5	17.6	6.2	3.6	8.6	47.6	

Annualized inputs to the CWD pool (total stem mass of dying trees divided by length of measurement interval) averaged 1.9–3.2 Mg/ha/yr, depending on plot group, for the entire 47 year period (Fig. 8). Fluctuations among measurement intervals can be attributed to the effects of occasional large-tree mortality in relatively small sample areas. However, both mixed and Tsuga- dominated fire group showed large increases in CWD input in the second half of the study period, from 1989 to 2009, compared to 1962–1989 (Fig. 7 and Table 3). Rates of CWD input nearly quadrupled over the last two decades, compared to earlier decades, for the ‘burn’ plot group.

Figure 8 Estimated CWD inputs, 1962–2009.

Annualized coarse woody debris (CWD) inputs, estimated as bole mass of trees dying during each measurement period, are generally higher during the last two decades than for the first 27 years of the study. No systematic differences in trend are evident among plot groups. Mass of trees dying during each measurement period is allocated uniformly across years within the measurement period.

Table 3 Coarse woody debris input estimates by plot group and period.

	CWD input (Mg/ha/yr)	
Year	1962–67	1967–89	1989–94	1994–99	1999–2004	2004–2009	Overall	1962–1989	1989–2009	
Acer (n = 5)	7.21	2.44	4.03	3.60	2.33	2.33	3.21	3.32	3.10	
Tsuga (n = 2)	5.78	0.92	1.82	0.31	3.77	2.03	1.89	1.82	1.96	
Mixed (n = 9)	0.95	2.04	1.73	3.87	5.11	3.68	2.59	1.84	3.40	
Burn (n = 3)	1.18	0.72	2.68	2.05	4.30	4.09	1.86	0.81	3.00	

Average annual CWD input was 13% (by mass) of total CWD in decay classes 1 and 2 in 2007. For plot groups, corresponding values ranged from 6% (‘burn’ group) to 20% (‘Acer’ group). These proportions are likely somewhat underestimated because input estimates do not incorporate large branches that were included in the CWD pool as measured in 2007.

Discussion

Aboveground biomass

Total aboveground woody biomass estimates for the Huron Mt. stands are among the highest obtained for eastern North American forests. Other estimates for old-growth hemlock-northern hardwood forests in the upper Great Lakes range from 200 to 325 Mg/ha (Mroz et al., 1985; Morrison, 1990; Rutkowski & Stottlemyer, 1993); all plot groups in this study exceed this range, with maximum values, in the Acer and Tsuga groups, at 430 Mg/ha, exceeding the highest previous estimates by a third. These values are comparable to estimates for southern Appalachian cover forests (Busing, 1993) although temperate coniferous forests in the Pacific northwest attain significantly higher biomass densities (e.g., up to 500 Mg/ha in Janisch, 2001). These high values could be, in part, an artifact of original plot selection in 1962, which might have favored stands with more undisturbed appearance, while cited studies might be based on less biased site selection or on larger areas of stands encompassing a range of disturbance. However, Lake Superior coastal climates are distinctive, and mesic forests near the lakeshore appear to experience unusually low frequencies of severe disturbance (Frelich & Lorimer, 1991), permitting high living biomass accumulation.

There were no clear, overall trends in biomass density over the full 47 years of the record; rather, biomass dynamics were dominated by shorter-term fluctuations (Fig. 3). For example, Acer and, to a lesser extent, Tsuga groups showed sharp decreases in biomass between 1962 and 1967, while all groups increased in biomass density during the long measurement interval from 1967 and 1989. Biomass increases in the first half of the study were proportionally greatest in the ‘burn’ group, and there was virtually no mortality in this group (Woods, 2000), suggesting that, 150 years after stand initiation, the stand remained in an aggrading phase. All groups show both increases and decreases in the four 5-year intervals since 1989, and trends are not synchronous among groups or stands. This pattern suggests biomass dynamics dominated by generally modest but irregular mortality of large trees superimposed on biomass gains through growth of surviving individuals. After 1990 (about 160 years after stand initiation), the ‘burn’ stand no longer shows an aggrading trend.

However, even though strong trends were not evident in total living biomass, directional changes are evident in forest structure, with an approximate doubling of proportions of biomass in trees >70 cm dbh (Figs. 4A and 5). Higher proportions (ca. 30%) of biomass in stems >70 cm dbh were observed by Brown, Schroeder & Birdsey (1997) in old forests of the lower midwest, but the largest stems in many of these stands were Quercus spp, while intermediate size-classes were dominated by Acer saccharum and Fagus grandifolia, suggesting that large Quercus stems were remnants of forest structures established under past, fire-dominated disturbance regimes. Tyrrell & Crow (1994) suggest that, in Tsuga- dominated forests of the upper Great Lakes, significant representation of stems >70 cm dbh indicates stand ages >275–300 yr. Here, biomass proportions in this size category approached or exceeded 20% by 2009 for Acer and Tsuga stands. The initial presence (and subsequent mortality without replacement—K Woods, 1989–2009, unpublished data) of large individual trees of shade-intolerant species (Quercus rubra, Pinus strobus, Populus grandidentata) in the ‘mixed slope’ group may be the signature of significant disturbance, but cores of late-successional canopy trees >250 years old suggest that such disturbances were either partial or ca. 300 years ago.

Earlier analysis of compositional trends in non-burned plots over the first 32 years of the study suggested gradual competitive displacement of mid-tolerant species (Betula alleghaniensis, Tilia americana) with increasing dominance of Acer saccharum or Tsuga canadensis, depending on soil properties (Woods, 2000). Biomass trends described here are consistent with this picture for the full 47 years of the data-set. In all plot groups, proportional representation of Acer and Tsuga has increased at the expense of all other species (Fig. 6). This is particularly evident in the ‘burn’ group, where proportional representation of Tsuga in the living biomass pool has increased continuously, particularly in the last two decades when total biomass has not been increasing consistently.

Taken together, these trends are consistent with the interpretation that old-growth hemlock-northern hardwood stands are typically in varying stages of long-term response to intermediate (or, in the case of the ‘burn’ plot-group and some individual plots, severe) disturbance (Abrams & Orwig, 1996; Woods, 2004; Woods, 2007; Bouchard, Kneeshaw & Bergeron, 2006; Stueve et al., 2011). Most notably, in terms of biomass dynamics, an approximate equilibrium in total biomass density is maintained while both size distributions and species representation show directional shifts.

Coarse woody debris

Overall, 2007 CWD volume (averaging 151 m3/ha for all plots) and biomass (46 Mg/ha), with about 25% of these values in standing snags, are within the range of earlier findings for old-growth forests of the upper Great Lakes and northeastern forest regions (Hura & Crow; Tyrrell & Crow, 1994; Goodburn & Lorimer, 1998; McGee, Leopold & Nyland, 1999; Angers et al., 2005; D’Amato, Orwig & Foster, 2008). Conifer CWD generally shows lower decay rates than hardwood CWD of comparable size (Harmon et al., 1986). However, studies comparing total CWD pools between hardwood and conifer-dominated forests have shown inconsistent results. Most of these, however, have focused on successional or boreal forests (Lee et al., 1997; Pedlar et al., 2002). Here, the ‘Tsuga’ plot group had relatively low CWD densities compared to other groups in this study, while the strongly hardwood-dominated ‘Acer’ group had very high values, suggesting substantially higher recent input rates for hardwood stands. Tyrrell & Crow (1994) show CWD increasing with age in Tsuga stands; values here are in the highest range for their study (suggesting stand ages >300 yr). CWD densities for the Acer group are among the highest published for cool-temperate deciduous forests. These pools are dominated by wood in the most decayed classes, possibly suggesting results of a significant, landscape-scale disturbance several decades ago; this may be consistent with the sharp decrease in biomass observed for these plots between 1962 and 1967.

Inputs of CWD from 1962–2009 averaged 2.5 Mg/ha/yr over all plots—equivalent to 5.5% of the 2007 total CWD pool and 11.7% of the pool of decay classes 1 and 2. If CWD pools are relatively stable or equilibrial, this implies an average residence time of around two decades overall and one decade in the relatively intact classes 1 and 2. Few studies have attempted to estimate residence time for CWD in hemlock-northern hardwood forests, and none have used long-term monitoring of mortality to assess input rates, so it is difficult to draw comparisons. However, existing estimates of residence time vary substantially. Zielonka (2006) estimates residence times of 13 and 24 yrs in classes 1 and 2 in conifer-dominated forests in Poland— substantially longer than turnover times suggested here under steady-state assumptions. However, estimates from old-growth hardwood forest in Ontario suggest a half-life for ‘down woody debris’ (exclusive of snags) of ca. 20 yr, with mean age of wood in classes 1 and 2 of three and five years (Vanderwel et al., 2008), and CWD in a logged, successional forest in New Hampshire decreased in mass by 90% in 20 years (Arthur, Tritton & Fahey, 1993). Given variations in approach (inclusion of snag mass or not, differences in decay class definitions) and coarseness of estimates, these decay rates could be compatible with current CWD pools being in approximately steady-state.

Nearly all previous studies of CWD dynamics are based on one-time measurements and space-for-time substitutions; consequently conclusions about trends over time are strongly assumption-laden. Direct measurement of trends in CWD input, based on long-term permanent-plots with multiple remeasurements, permits an unusual direct assessment of trends. Most strikingly, overall per-year CWD input increased by nearly half between the first 27 years of the study and the last 20 years (from 2.1 to 3.1 Mg/ha/yr), suggesting non-equilibrial status of biomass and carbon dynamics at decadal scales. Such trends might be anticipated in cohort-structured stands where one or more cohorts are reaching senescence; consistent with this interpretation, overall canopy-tree mortality rate increased from 0.6%/yr for 1962–1989 to 0.9%/yr from 1989–2009. These increases were most evident in the youngest stands—the ‘burn’ group—with an increase in CWD input from 0.8 to 3.0 Mg/ha/yr for these two periods, and in the mixed plots (increase 2.1 to 3.9 Mg/ha/yr).

Over all plots, size distribution of stems contributing to total mortality and CWD input did not show the increasing dominance of large stems seen in the living biomass pool (Fig. 4B), but size distributions of CWD input do suggest an increased importance of smaller to mid-size canopy trees (ca. 25–50 cm dbh) in the last two decades. This may be due to enhanced mortality of suppressed sub-canopy trees and sub-dominant canopy trees.

The ‘Acer’ group, with the highest CWD inputs in 1962–1989, and the ‘Tsuga’ group showed less change; if these two groups are in approximate steady-state, implied residence time of CWD is about 16 years overall and 5 years in decay classes 1 and 2 for the hardwood-dominated ‘Acer’ plots, and about 22 years and 10 years for the ‘Tsuga’ group. This is in keeping with general expectations of more rapid decay of hardwood CWD and, for Acer plots, consistent with estimates from Liu et al. (2006) and Gough et al. (2007), who estimate a decomposition rate-constant of 0.09/yr, implying residence time of 11–13 yrs (Liu et al., 2006), for forests of similar composition. Arthur, Tritton & Fahey (1993) estimate 90% loss of CWD biomass in 23 yr for hardwoods in New Hampshire; this would be a more advanced state of decay than in the classes incorporated in this study. However, calculations based on steady-state assumptions should be approached cautiously; 47 years is much less than half of canopy turnover time estimates for similar forests (Frelich & Graumlich, 1994; Dahir & Lorimer, 1996).

However, several factors make these interpretations very tentative. First, input estimates here may be low because of exclusion of branch biomass for dying trees. Second, changes in size distribution towards increasing dominance of very large stems suggest that CWD inputs will tend to shift to larger-diameter materials, with correspondingly longer residence-times. Finally, over longer periods, canopy-tree mortality may be dominated by pulses associated with more severe disturbance (Woods, 2004; Woods, 2007; Frelich & Lorimer, 1991). No such events have occurred during the study period or, as suggested by size and age structures (Woods, 2000), in several decades prior to its initiation. Thus long-term average inputs are likely to be greater than those observed here. All of these factors suggest that estimates of recent CWD input for the study plots may be an underestimate of longer-term, future inputs. If so, CWD pools may be expected to increase.

Sources of error and uncertainty

While the long-term data-set suppporting these analyses supports unusually detailed insight into carbon/biomass dynamics and relationships between tree growth and mortality and CWD pools in old-growth forests, some important uncertainties and potential errors remain. Two of these are likely most important.

First there are large uncertainties connected with biomass estimation for very large trees (>70 cm dbh) that constitute an increasing proportion (up to 30%) of total above-ground living biomass. Until allometric relationships for large stems become available, it will be difficult to reduce this uncertainty. All estimates of biomass in late-successional forests are subject to the same potential errors.

Second, even though the study period of 47 years is unusually long, canopy-tree mortality rates of <1%/yr indicate that it remains brief in the innate time-scale for dynamics in these forests. Observed dynamics represent less than half of a single generation of canopy trees. Further, multiple studies have suggested that dynamics in similar forests are strongly influenced by major disturbance episodes with return times measured in centuries (Woods, 2004; Woods, 2007; Frelich & Lorimer, 1991). Such events would have major consequences for CWD dynamics, but none have occurred at this site during the study period.

Conclusions

These results provide at least partial answers to my four initial questions; all can be more confidently addressed only with longer-term observation:

Do living biomass pools show consistent trends at decadal time scales in these old-growth, late-successional forests? Overall, biomass densities observed here are exceptionally high in the context of published estimates for northeastern North America. However biomass estimates from seven measurements over nearly five decades indicate no consistent long-term trend. Live biomass pools are neither aggrading or declining overall, but most stands show significant fluctuations at a decadal scale. Three plots in an even-aged, post-fire stand appeared to be aggrading for the first 25 years of the study, but this trend ended about 1990 (stand age ca. 160 yr). These results do not suggest that above-ground living biomass pools are significant carbon sinks, but longer-term studies will be required to make this conclusion with high confidence. Consistent long-term increases in the contribution of very large trees to living biomass indicate non-equilibrial demography and the potential for sharp biomass fluctuations in future decades with mortality of these trees.

What are patterns of CWD input through tree mortality at decadal scales? Fluctuation in CWD input rates are large, and individual plots are strongly influenced by occasional mortality of large canopy trees. However, in nearly all plot groups, input rates increased significantly from the first 27 years of the study to the last 20 years. This pattern is consistent with the suggestion from previous analyses (Woods, 2000) that stand dynamics are influenced by episodic events translated through tree demography; in terms of size-structure and species composition these stands may be cohort-structured and non-equilibrial.

Are historical CWD input rates and current CWD pools consistent with a steady-state assumption regarding CWD reserves? Estimates of CWD input and standing volume here do not allow confident projections regarding wheter CWD pools are stable, increasing, or decreasing. However, increasing dominance of large canopy trees (with the longer residence time of larger tree boles in the CWD pool) and the absence of major disturances during the study period to date may be consistent with an expectation of long-term increases in carbon pools in dead biomass with future mortality of these large trees, even if living biomass is not increasing.

How are forest composition and history related to biomass and carbon dynamics? While total CWD density did not show marked variation among stand types, Tsuga-dominated stands exhibited consistently lower input rates. High fluctuation in CWD input rates may be consistent with cohort-structured mortality related to effects of rare disturbances on tree establishment rates (Woods, 2000; Woods, 2004; Woods, 2007), particularly in hardwood-dominated stands.

Supplemental Information

Table S1 Sources for allometric equations

Click here for additional data file.

Table S2 Biomass density estimates, all plots and species

Click here for additional data file.

Table S3 CWD by plot, species, and decay class

Click here for additional data file.

Study plots were established by Eric Bourdo of the Ford Forestry Center, Michigan Technological University; information about plot location and data from 1962 and 1967 measurements were generously shared by Dr. Bourdo and the staff at the Ford Forestry Center. Approximately 25 Bennington College undergraduate students have participated in field measurements since 1989. The Huron Mt. Wildlife Foundation has provided logistical support through the facilities of the Ives Lake Research Station.

Additional Information and Declarations

Competing Interests

Author Contributions

Data Deposition

The author declares there are no competing interests.

Kerry D. Woods conceived and designed the experiments, performed the experiments, analyzed the data, contributed reagents/materials/analysis tools, wrote the paper, prepared figures and/or tables, reviewed drafts of the paper.

The following information was supplied regarding the deposition of related data:

Knowledge Network for Biocomplexit: DOI 10.5063/F1PC3085.

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
