# Peer review of "Multi-decade biomass dynamics in an old-growth hemlock-northern hardwood forest, Michigan, USA"

_PeerJ, doi:10.7717/peerj.598_

## Round 0.1 · original submission · Minor Revisions

· Academic Editor

Minor Revisions

Both of your reviewer's were felt your paper is a valuable contribution. There are some useful suggestions that your should consider and respond to as appropriate. I found your use of first-person was distracting and reduced this reader's focus on the research. This is your choice but you might consider this in your revision.

·

Basic reporting

This paper is nicely constructed, with clear objectives, well articulated presentation of prior research, easily interpreted findings, and informative interpretation and discussion. My only substantive criticism is that Tables 1-3 repeat data presented in Figures 3, 7 and 8. Such repetition makes it easier for subsequent users to extract data, but if space is limiting the tables could be eliminated.

Experimental design

This is an observational study based on repeated sampling of plots over 47 years. The author has quite reasonably decided to group subsets of these plots based on qualitative floristic similarity such that the number of ‘replicates’ within a plot group is quite small (n=2-9). Nonetheless, the statistics applied are appropriate and the data are not over-interpreted. As importantly, sources of error and uncertainty resulting from the use of allometric scaling equations of unknown accuracy are accounted for and discussed.

Validity of the findings

This is a unique data set chronicling multi-decadal biomass and coarse woody debris change in a cool temperate eastern old-growth forest and is important in and of itself. The author frames his paper with four pertinent questions regarding the ecology of these systems and presents well-reasoned conclusions as to the extent to which firm answers to these questions can be reached, based on the data. Findings are supported by the data and advance our understanding of these rare ecosystems.

Reviewer 2 ·

Basic reporting

No comments

Experimental design

No comments

Validity of the findings

No comments

Additional comments

I found the manuscript of considerable interest and found it to be technically sound. The quality of the data and length of the study are truly noteworthy. The only limitation is the lack of published biomass equations for large trees, but the author is appropriately cautious and discusses this at length in both the methods and discussion.
Specific comments:
Abstract, line 14 of text. Change “have are neither increasing nor decreasing” to “have neither increased nor decreased”
Body of manuscript
L50. Insert “varying” after “history”
L139. Replace “plog” with “plot”
L150. What was the statistical approach? It should be described in the methods rather than in the results section.
L156. Change “vales” to “values”
L161. Change “is” to “was”
L197-198. Awkward, please clarify.
L228. Change “70%” to “70 cm dbh”?
L263. Change “hoever” to “however”
L306-312. This section would be improved by adding some references and values for decay rates of the major species. A residence time of 16 years seems short in comparison to other studies.
L322. Delete extra carriage return.
L367-368. Please revise sentence for clarity.

---

## Round 0.2 · accepted · Accept

· Academic Editor

Accept

Hello Kerry,

Thanks for your changes. I think it is a very nice contribution. It's been a long, long time since our paths have crossed. I hope all is well.

Hank